# Implications of diffusion and time-varying morphogen gradients for the dynamic positioning and precision of bistable gene expression boundaries

Melinda Liu Perkins ◉ *

Developmental Biology Unit, European Molecular Biology Laboratory, Heidelberg, Germany

* mindy.liu.perkins@embl.de

**Data Availability Statement:** Code to generate all simulations in the text is available as a supplementary file.

## Abstract

The earliest models for how morphogen gradients guide embryonic patterning failed to account for experimental observations of temporal refinement in gene expression domains. Following theoretical and experimental work in this area, dynamic positional information has emerged as a conceptual framework to discuss how cells process spatiotemporal inputs into downstream patterns. Here, we show that diffusion determines the mathematical means by which bistable gene expression boundaries shift over time, and therefore how cells interpret positional information conferred from morphogen concentration. First, we introduce a metric for assessing reproducibility in boundary placement or precision in systems where gene products do not diffuse, but where morphogen concentrations are permitted to change in time. We show that the dynamics of the gradient affect the sensitivity of the final pattern to variation in initial conditions, with slower gradients reducing the sensitivity. Second, we allow gene products to diffuse and consider gene expression boundaries as propagating wavefronts with velocity modulated by local morphogen concentration. We harness this perspective to approximate a PDE model as an ODE that captures the position of the boundary in time, and demonstrate the approach with a preexisting model for Hunchback patterning in fruit fly embryos. We then propose a design that employs antiparallel morphogen gradients to achieve accurate boundary placement that is robust to scaling. Throughout our work we draw attention to tradeoffs among initial conditions, boundary positioning, and the relative timescales of network and gradient evolution. We conclude by suggesting that mathematical theory should serve to clarify not just our quantitative, but also our intuitive understanding of patterning processes.

## Author Summary

In many developmental systems, cells interpret spatial gradients of chemical morphogens to produce gene expression boundaries in exact positions. The simplest mathematical models for positional information rely on threshold detection, but such models are not

**Funding:** M.L.P. was supported by a fellowship awarded through the EMBL Interdisciplinary Postdoc Programme EIPOD4, which is co-funded by the European Molecular Biology Laboratory and Marie-Skłodowska Curie Actions (grant agreement number 847543). https://www.embl.de/training/postdocs/08-eipod/EIPOD4-programme/ https://ec.europa.eu/research/mariecurieactions/node_en M.L.P. received a graduate student researcher salary through Air Force Office of Scientific Research grant FA9550-18-1-0253 awarded to Murat Arcak. https://www.afrl.af.mil/AFOSR/ The funders had no role in study design, data collection and analysis, decision to publish, or preparation of the manuscript.

**Competing interests:** The author has declared that no competing interests exist.

robust to variations in the morphogen gradient or initial protein concentrations. Furthermore, these models fail to account for experimental results showing dynamic shifts in boundary placement and increased boundary precision over time. Here, we argue that dynamic positional information is interpreted differently by a bistable patterning system depending upon whether gene expression products diffuse. We explore two mathematical methods to analyze pattern refinement with and without diffusion, and discuss design tradeoffs among precision, placement, accuracy, and timescale. We suggest that future research into dynamic positional information would benefit from perspectives that link local (cellular) and global (patterning) behaviors, as well as from mathematical theory that builds our intuitive understanding alongside more data-driven approaches.

## Introduction

Spatial patterning of gene expression, especially during embryonic development, has fascinated theoretical and experimental researchers throughout the past century. A central question involves how an initial pattern is interpreted by cells in a tissue to produce sharp boundaries between spatial domains expressing distinct sets of genes [1, 2]. Significant evidence indicates that concentration gradients of chemical morphogens provide the necessary *positional information* for accurate boundary formation in various contexts ranging from axis orientation [3–6] to neurogenesis [7, 8]. In early studies, cells were proposed to interpret positional information through a thresholding mechanism, whereby a cell expresses a certain gene only if the morphogen concentration in that cell exceeds a certain level [9]. Subsequent research has complicated this hypothesis, with experimental evidence indicating that features such as the slope or temporal evolution of a morphogen gradient may also transfer positional information to target cells [10–16].

The recent concept of *dynamic positional information* posits that positional information is not conferred through a static reading out of morphogen concentration, but rather through a dynamic process of interplay between morphogen concentration and the genetic networks that interpret it [17]. Dynamic positional information has been inspired in part by two observations. First, the precision (regularity) of boundaries increases over time, which in combination with domain shifting leads to evolution of patterns from an initial messy form to a refined final form that cannot be directly predicted from the concentrations of underlying signals [18–20]. Second, gene expression domains shift over time, as indicated by experimental evidence from fruit flies [18, 21–25], scuttle flies [26], and vertebrate neural tube [27]. To explain these observed behaviors, researchers have investigated stochastic influences [28, 29], the structure of biochemical networks [30], protein diffusion [20], biomechanical forces including cell sorting or aggregation [31–33], and the temporal evolution of upstream patterns [15].

Dynamical systems analysis—in which differential equations describe how chemical concentrations evolve in time—underlies most efforts to understand dynamic positional information. In a dynamical systems model, morphogen concentration typically acts as a spatially varying bifurcation parameter that modulates the number and stability of the steady states attainable by a genetic network (and therefore by a cell located at a particular point in space). The morphogen modulates gene expression rates such that cells destined for one side of a gene expression boundary favor one stable state and cells on the other side favor another stable state. Many models neglect the diffusion of gene expression products so as to simplify simulation and analysis without completely compromising qualitative theoretical-to-empirical fit [15, 21, 30, 34, 35]. However, diffusion is known to be a potential determinant of

patterning in chemical and biological systems alike—as demonstrated by Turing's famous observation [36]—and theoretical and experimental research supports the idea that such intercellular signaling may reduce noisy influences or correct patterning defects in developmental systems [20, 37, 38].

While many studies focus on specific experimental systems, some recent research into dynamical positional information has leveraged a bottom-up approach by examining a simple, generic model—the bistable genetic network (toggle switch)—and generalizing the conceptual or mathematical insights thereafter. For example, the authors of [39] classified the qualitative behaviors of attractors and trajectories in toggle switches with time-dependent (morphogen) inputs and subsequently used these classifications to describe the dynamic behavior of the gap gene network in fruit flies [15]. Similarly, the authors of [29] found that stochastic gene expression could result in boundary refinement over specific timescales, and confirmed the effect experimentally in a larger network [27]. Such approaches leverage a thorough understanding of simple models to make sense of more complex cases, enabling researchers to generate new insights, predictions, and perspectives along the way.

Here, we use a simple deterministic bistable network to exemplify how the diffusion of gene expression products determines the mathematical mechanisms behind transient boundary positioning, and therefore how cells may be considered to interpret positional information from morphogen concentration. Without diffusion, morphogen concentration drives individual cell trajectories directly, such that boundaries shift indirectly as a result of many coordinated but isolated behaviors considered together (e.g., [29]). With diffusion, local morphogen concentrations alone do not dictate individual cell behaviors, but can instead be understood to determine the speed of movement of the entire gene expression boundary in a predictable way. For each case we suggest a scheme to improve the reproducibility of boundary placement when initial conditions may vary across embryos; in the system without diffusion, this scheme is identical to one that improves the precision of a boundary when initial conditions vary among cells. We are guided throughout by the belief that exploring the constraints and affordances of hypothetical designs can facilitate our understanding of extant living systems as well as our ability to engineer them.

The paper is broken into two complementary parts. The first part concerns systems where gene products do not diffuse and in which morphogen gradients emerge or disappear over time. As a consequence of morphogen dynamics, bifurcation points across the tissue shift in space, which influences both the transient placement of boundaries and the precision (in a single embryo) or reproducibility (across embryos) of a final pattern. We introduce a metric for assessing boundary precision (reproducibility) as the width of the region in which cells receiving the same concentration of morphogen do not approach the same steady state (adopt the same fate). We use the metric to analyze a design for enhancing precision during or after boundary formation. This design relies on temporal decrease in morphogen concentration to sweep cells in imprecise bistable regions into monostable regions where gene expression is uniform for cells sharing the same morphogen concentration. Central to this discussion is the role of initial conditions and the relative rates of network and gradient evolution in determining precision, with relatively slower gradients and longer times to readout generally producing more precise outcomes.

In systems where gene products diffuse, an individual cell's dynamics are no longer determined solely by its morphogen concentration. In the second part of this paper, we analyze diffusion-driven boundary shifting by introducing a mathematical approximation for predicting the propagation of a generic bistable wavefront when bifurcation parameters vary in space. This approximation allows us to reduce a full PDE simulation to the simulation of a single-variable ODE once fronts have formed in bistable regions. We apply this approximation to

biological patterning systems in which morphogen concentration determines the local propagation velocity of a gene expression boundary. We argue the perspective is relevant to current thinking about patterning by examining an existing model for the formation of the Hunchback (Hb) boundary in early *Drosophila* [40], which exhibits both spatial and temporal variation in the underlying Bicoid (Bcd) morphogen gradient. We note constraints such as the transience of the pattern and the issue of scaling in the absence of corrective mechanisms to the gradient or initial conditions. Next, we show how an understanding of propagating wavefronts enables us to develop new designs, including a robust mechanism for accurate steady-state boundary placement even from homogeneous initial conditions.

Our analysis throughout this work highlights how temporally varying inputs and cell-to-cell coupling can regulate design tradeoffs among boundary precision, accuracy of placement, and the necessity for accurate initial conditions. We propose a subtle shift in focus from individual cell trajectories to the evolution of entire gene expression boundaries in order to derive a more complete picture for the regulation and interpretation of dynamic positional information on local and global scales. Most importantly, we aim to illustrate how a change in perspective can facilitate intuitive understanding of bistable patterning systems, which may inspire further thinking about potential designs in natural contexts and for synthetic applications.

## Results

### Boundary positioning and precision: Reliance on initial conditions

The simplest models for dynamic patterning assume static morphogen gradients. Experimentally, however, these gradients have been observed to change in time [18, 41–43]. In particular, temporal decreases in gradient amplitude have been suggested to play an essential role in properly positioning gene expression boundaries [15]. Here, we show that time-varying morphogen gradients may not only modulate boundary position, but simultaneously increase robustness of patterning outcomes to variation in initial conditions. First, we introduce a metric to assess boundary precision or reproducibility in placement in the context of an emerging morphogen gradient acting simultaneously with the genetic network. We next apply the metric to hypothesize that allowing morphogen concentrations in an existing gradient to decrease over time could enhance boundary precision (reproducibility) by increasing robustness to variation in initial conditions that would otherwise cause neighboring trajectories to diverge to different stable steady states. In both cases we note the design tradeoffs among precision, time to read-out, and the relative rates of gene network and morphogen dynamics.

In this and the following section, we will approximate a tissue as a continuum, with dynamical behaviors varying along a coordinate $x$. For the sake of discussion we will take $x$ to be the anterior-posterior axis of a model embryo. Assuming for now that parameters are static (not time-varying), we consider chemical reactions (without diffusion) governed by equations of the form

$$\frac{du(t,x)}{dt} = f(x, u(t,x)) \tag{1}$$

where $u(t,x)$ are chemical concentrations at time $t$ and coordinate $x$, and $f$ incorporates the production and degradation rates of the chemicals. The whole system is presumed to be mono- or bistable at all points on the axis. For example, in this section we will consider a chemical toggle switch in which there are two mutually repressive species $u_1$ and $u_2$. The species $u_1$ is activated by morphogen $\alpha$, which is expressed in a gradient along $x$ with length

constant $\lambda$:

$$\alpha(x) = be^{-\frac{x}{\lambda}}. \tag{2}$$

For simplicity, we also lump transcription and translation into a single step. The governing equations are then

$$\begin{cases} \frac{du_1(t,x)}{dt} = a_1 h_a(\alpha(x)) h_r(u_2(t,x)) - \beta u_1(t,x) \\ \frac{du_2(t,x)}{dt} = a_2 h_r(u_1(t,x)) - \beta u_2(t,x) \end{cases} \tag{3}$$

where

$$h_a(y) = \frac{\left(\frac{y}{K_a}\right)^{n_a}}{1 + \left(\frac{y}{K_a}\right)^{n_a}}, \quad h_r(y) = \frac{1}{1 + \left(\frac{y}{K_r}\right)^{n_r}} \tag{4}$$

are activating and repressing Hill functions with coefficients $n_a$, $n_r$ and concentration to half maximum $K_a$, $K_r$ respectively.

To consider morphogen gradients that are not static, we modify the governing equation Eq (1) to be time dependent:

$$\frac{du(t,x)}{dt} = g(t, x, u(t,x)). \tag{5}$$

The same genetic network can thus respond differently to the same concentrations of $u_1$ and $u_2$ depending upon the $x$-coordinate at which the cell is located and the time at which the concentrations are measured. We refer to cell behavior at a particular location $x$ as *local* and at a particular time $t$ as *instantaneous*.

An instantaneous phase portrait corresponding to a fixed time $t$ is derived from the equation

$$\frac{d\tilde{u}(\tau, x)}{d\tau} = g(t, x, \tilde{u}(\tau, x)), \tag{6}$$

where the only difference from Eq (5) is that $g$ is fixed in time. For our purposes, this equation is derived by taking the morphogen profile at a time $t$ from Eq (5) and creating a new system in which the morphogen is static with that profile for all times $\tau$, the integration variable in the hypothetical system (6). For a more general discussion of instantaneous phase portraits, see [15, 39].

Although we will use a morphogen gradient as an example, we note that the intuitive ideas discussed here apply to other spatial gradients that are monotonic but not necessarily exponentially decaying in space (e.g., gene expression boundaries) or that evolve monotonically but not necessarily exponentially in time.

**Boundary formation during gradient emergence.** We begin by investigating boundary placement and precision in a network with a morphogen gradient $\alpha(t, x)$ that grows from 0 at $t = 0$ to a final profile $\alpha(t \to \infty, x) = \alpha_\infty(x) = be^{-\frac{x}{\lambda}}$ with length constant $\lambda$, modeled as

$$\alpha(t, x) = be^{-\frac{x}{\lambda}}(1 - e^{-\gamma t}). \tag{7}$$

The conclusions discussed below generalize to other functional forms, but we choose an exponential to make $\gamma$, the rate of approach to the final profile, explicit.

Suppose that initial conditions are uniform (identical across all cells). At each point $x$ along the embryo, the initial conditions fall into the basin of attraction of high $u_1^*$, low $u_2^*$ (to the

anterior) or low $u_1^*$, high $u_2^*$ (to the posterior). The location of the boundary is determined by the coordinate $x^*$ where the initial conditions cross from one basin of attraction to the other. Embryo-to-embryo variability in the morphogen profile compromises the reproducibility of boundary placement by shifting $x^*$. Similarly, if individual embryos have uniform initial conditions but those conditions vary across embryos, then the position of the boundary may also vary by embryo even if the gradient is identical across embryos. The degree of variation depends on the embryo-to-embryo variability in initial conditions: If they all fall within the same basin of attraction across the entire bistable region, then boundary placement will be identical across embryos. Otherwise boundary placement will vary across some width less than or equal to the width of the bistable region, based on the spread in initial conditions.

Mathematically, the question of how much embryo-to-embryo variability in initial conditions propagates to variability in boundary placement is similar to the question of boundary precision within a single two-dimensional embryo where initial conditions vary across cells at the same coordinates on the A-P axis. Such variation in initial conditions despite deterministic governing dynamics may emerge, for example, when the network of interest is more tightly regulated than preceding processes. A precise boundary is achieved when initial conditions for all cells with coordinates $x < x^*$ fall within the basin of attraction of the state with high $u_1^*$ and all those with coordinates $x > x^*$ fall within the basin of attraction of the state with low $u_1^*$ (and $0 < x^* < L$, the embryo length). If the system is everywhere monostable at $\alpha_\infty(x)$ and transitions from high to low $u_1^*$ with increasing $x$, the boundary will be precise regardless of initial conditions though the boundary may not be very sharp. Otherwise, some region of the embryo is bistable as $t \to \infty$, and precision is determined by the relationship between initial conditions and the basins of attraction for the two states in this region (Fig 1).

Let $x_a$ mark the anterior end of the bistable region and $x_p$ mark the posterior end as $t \to \infty$. We denote the basin of attraction for high $u_1^*$ at coordinate $x$ by $\mathcal{B}_x$. For the gradient (7), $x_1 < x_2$ implies $\alpha(t, x_1) \geq \alpha(t, x_2)$ at all points $t$. Since higher concentrations of morphogen bias the system in favor of high $u_1^*$, any initial condition that can achieve high $u_1^*$ for a certain concentration of morphogen will also achieve that state for higher concentrations of morphogen; i.e., all initial conditions ending in high $u_1^*$ at $x_2$ will also end in high $u_1^*$ for $x_1$, but not vice versa. In other words, $\mathcal{B}_{x_2}$ is a subset of $\mathcal{B}_{x_1}$.

Assume we are given a set of initial conditions not knowing how they may be distributed across $x$. If there exists an $\tilde{x}_a$ between $x_a$ and $x_p$ such that $\mathcal{B}_{\tilde{x}_a}$ contains *all* the initial conditions, then the maximum such $\tilde{x}_a$ determines the guaranteed boundary between the region of high $u_1^*$ and a possibly irregular region. Similarly, if there exists an $\tilde{x}_p$ between $x_a$ and $x_p$ such that $\mathcal{B}_{\tilde{x}_p}$ contains *none* of the initial conditions, then the minimum such $\tilde{x}_p$ determines the guaranteed boundary between the irregular region and the region of low $u_1^*$. Thus, the ratio

$$R = \frac{\tilde{x}_p - \tilde{x}_a}{x_p - x_a} \tag{8}$$

captures the fraction of the final bistable region which may be irregular, with $R = 0$ indicating no irregularity (boundary is precise and located at $\tilde{x}_a = \tilde{x}_p$). Otherwise, there are no guarantees for precision without further information about the spatial distribution of the initial conditions. We note that if instead of cell-to-cell variation in a single embryo, we took the initial conditions to represent embryo-to-embryo variation in uniform starting concentrations, $R$ is a measure of reproducibility that captures the fraction of the maximum possible variation in boundary placement that is actually observed among the embryos.

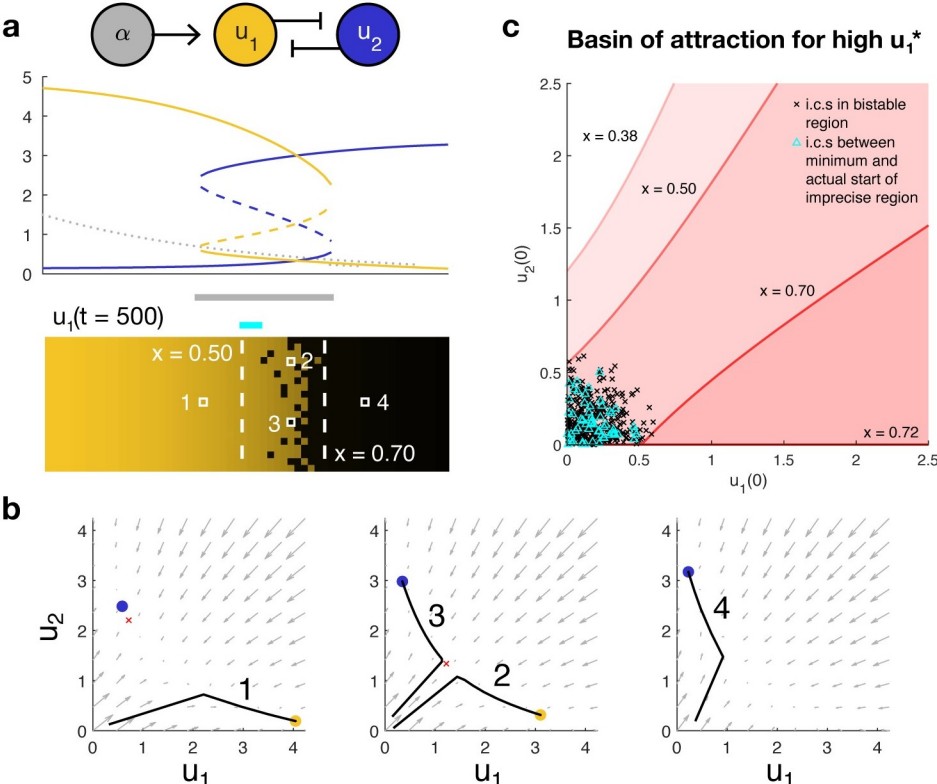

**Fig 1. (a) A bistable toggle switch can produce a gene expression boundary from a morphogen gradient**. Top, the morphogen (gray) induces expression of $u_1$ (gold), which is mutually repressive with species $u_2$ (blue). The morphogen concentration causes the anterior (left) of the boundary to be monostable in favor of high $u_1$ and the posterior (right) to be monostable in favor of high $u_2$. In the bistable region of the boundary itself, the initial conditions determine whether cells express high $u_1$ or high $u_2$ at steady state. Middle, a bifurcation diagram indicates the value of the steady states, with solid lines indicating stability and dashed lines, instability. Bottom, a heat map of $u_1$ concentration at steady state showcases how variation in initial conditions among rows may result in an imprecise boundary in the bistable region. **(b) Phase portraits at different $x$ coordinates along the embryo predict how trajectories evolve over time**. Gray arrows indicate the direction in which a trajectory will travel. Solid dots indicate stable steady states favoring high $u_1$ (gold) or high $u_2$ (blue). The unstable middle point in the bistable region is denoted by a red ×. Black lines are example trajectories corresponding to the cells labeled above. **(c) The boundaries of the imprecise region are determined by the basins of attraction that bound the initial conditions**. Red shaded regions beneath each red curve denote the sets of initial conditions that will converge to the state with high $u_1^*$ at a fixed coordinate $x$. Black × denote the initial conditions of all cells in the bistable region. The distance between the lowest and highest $x$ that most tightly bounds these conditions (here, $x_a = 0.50$ and $x_p = 0.70$) is the most conservative estimate of the width of the imprecise region. The actual distribution of initial conditions may cause the actual width of the region to be smaller; for example, here, the initial conditions for the anteriormost cells within the bistable region (cyan △) are upper bounded by a higher $x$ than are the black ×.

The basins of attraction depend crucially on the relative rates of gradient formation and gene network dynamics. As a general guidepost, we evaluate the relative rates using the ratio

$$\kappa = \frac{\gamma}{\sigma}, \tag{9}$$

where $\sigma$ is the minimum rate of convergence to a stable steady state across all bistable coordinates $x$ when $t \to \infty$ (see S1 Appendix). $\kappa < 1$ indicates that the dynamics of the network are generally faster than those of the gradient, while $\kappa > 1$ indicates the dynamics of the gradient are generally faster than those of the network. When $\kappa = 0$ we may assume the network equilibrates instantly to any changes in the gradient, so the trajectories traversed by the network are

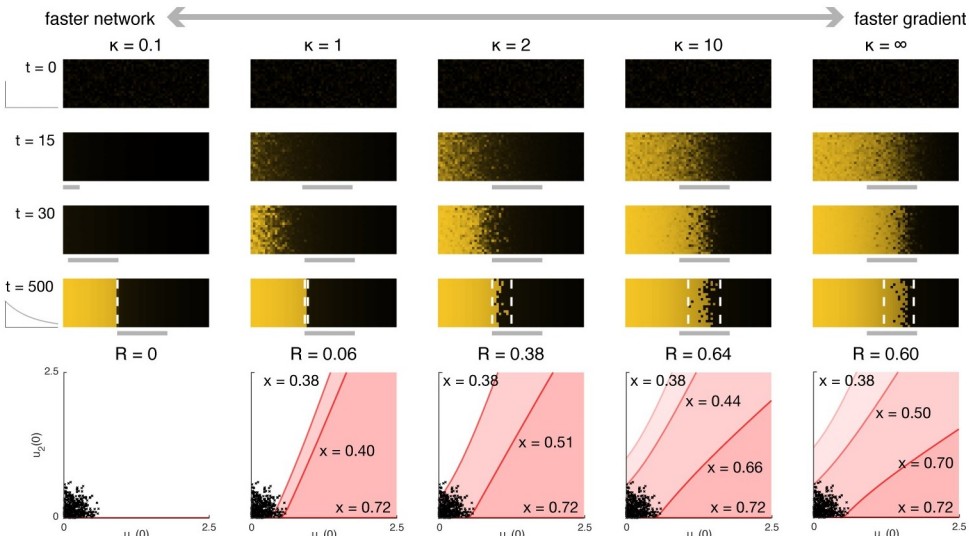

**Fig 2. The precision of a gene expression boundary forming atop a dynamically growing gradient depends on the relative rates at which the network and the gradient evolve.** The control (static gradient) corresponds to $\kappa = \infty$. Top, time lapses illustrating the concentration of $u_1$ across a whole embryo. Assuming the gradient emerges from a uniformly monostable state (here with $\alpha(t = 0, x) = 0$), then the slower the gradient emerges, the more precise the resulting boundary will be. Boundary formation lags the movement of the bistable region from anterior to posterior. Bottom, the width of the imprecise region can be predicted from the basins of attraction at each coordinate $x$; see Fig 1. $R$ quantifies the maximum width of the imprecise region (which depends on the initial conditions) relative to the width of the bistable region, with lower $R$ indicating higher precision. That $R$ is higher for $\kappa = 10$ relative to $\kappa = \infty$ is an artifact of the particular choice of initial conditions, which are shared across all simulations pictured here.

approximated by the trajectories of the instantaneous steady states. In the opposite case $\kappa \gg 1$, the dynamics of the genetic network are well approximated by those of an equivalent system with static gradient $\alpha_\infty(x)$. Note that if the system is monostable at $t = 0$, then $\kappa = 0$ guarantees precision in the resulting boundary. All other schemes depend on establishing proper initial conditions to achieve precision.

Fig 2 illustrates simulations for varying $\kappa$ (achieved by varying $\gamma$) with the same set of randomly generated initial conditions clustered near $(0, 0)$. At $\kappa \ll 1$, the network equilibrates to the single steady state available from $t = 0$ so that by the time the bistable region emerges the variation in initial conditions has effectively been washed out. All cells at the same coordinate $x$ then follow the same trajectory, resulting in a precise final boundary. As $\kappa$ increases, the imprecision ($R$) also tends to increase (the slight improvement in precision for $\kappa \to \infty$ relative to $\kappa = 10$ is an artifact of the particular set of initial conditions chosen for this example). Even with relatively slow gradient emergence, the formation of the boundary lags the movement of the bistable region and appears to track its movement from anterior to posterior. For moderate $\kappa$, the region of high $u_1^*$ expression becomes more uniform as bistable regions toward the anterior become monostable, causing initially divergent trajectories to redirect toward the same single steady state. The net effect is that precision (in the sense of similarity in expression levels among neighboring cells) qualitatively appears to increase over time across the anterior of the embryo. Hence, both boundary positioning and precision can result from the same underlying dynamic process of gradient formation.

Although we have considered here the case where a gradient grows from an initially homogeneous, monostable regime, there is no mathematical reason that $\alpha(t = 0, x)$ must be uniform, nor that the gradient itself must stabilize as $t \to \infty$. The next section will explore an example

in which a gradient already exists at time $t = 0$. A minor adjustment to the metric allows precision to be evaluated at finite $t$ and does not fundamentally depend upon whether morphogen concentration increases or decreases in time.

**Precision enhancement after gradient emergence.** We hypothesize that temporal decrease in morphogen concentration in an established gradient can increase precision (equivalently reproducibility in placement across embryos) by "sweeping away" boundary irregularities regardless of the prior process that produced them. To describe the intuition behind the idea, we will first assume that the gradient has remained static long enough for the genetic network in all cells to have reached a steady state. We assume that as we proceed along the $x$ coordinate, the system transitions from monostable favoring $u_1$ to bistable to monostable favoring $u_2$, as in Fig 1A. At time $t = 0$, the morphogen concentration begins to decrease exponentially in time, taking the form

$$\alpha(t, x) = b e^{-\frac{x}{\lambda} - \gamma t} \tag{10}$$

at coordinate $x$ and time $t$. Such temporal decrease might arise from increased degradation of morphogen or from a reduction in morphogen production, both of which have been observed for the Bcd gradient in fruit flies. The molecular mechanism could determine, for example, whether the morphogen profile stabilizes in the shape of a gradient with a lower concentration (as might be expected from increased degradation) or becomes completely flat (as might be expected from decreased production) [44]. Both cases may be handled in our framework, though for this example we have structured the gradient to approach 0 everywhere as $t \to \infty$.

Introducing temporal decrease in $\alpha$ causes the transitions between monostable and bistable regions to shift anteriorly (toward low $x$) over time, such that parts of the initially monostable region for high $u_1$ will become bistable, and part or whole of the initially bistable region will become monostable for low $u_1$. Cells with high $u_1$ at $t = 0$ remain at high $u_1$ even if the region where they are located becomes bistable. At the same time, the initially bistable region becomes monostable for low $u_1$, thereby "sweeping away" irregularities introduced by the initial bistability. The result is that an imprecise boundary can be made precise through the dynamics of the morphogen gradient (Fig 3A). Maximum precision is achieved once the initially irregular region has become entirely monostable. A side effect of this scheme is that the placement of the boundary, as determined by the locations of the bifurcation points, also shifts anteriorly. (Note that a similar effect could be observed but with a posterior shift if the morphogen concentration were increasing in time across the domain.).

Although we have assumed the genetic network in all cells has reached a steady state before time $t = 0$, intuitively we might expect that, if the temporal decrease in $\alpha$ is slow enough relative to system dynamics, trajectories in the anteriormost region could still converge to high $u_1^*$ even if they do not begin at that steady state. We might therefore ask for what set of initial conditions (at time $t = 0$) a trajectory at a given $x$ coordinate will converge to high $u_1^*$. Similarly, given an arbitrary set of initial conditions, we might ask how much irregularity we expect to persist despite decay in the gradient; that is, how much we expect temporal decrease in morphogen concentration to enhance precision relative to a control with no decrease.

To investigate these questions, we first note that in the current model the morphogen concentration will decay to zero and only the monostable state for high $u_2^*$ will remain. For the boundary to be maintained we require some mechanism to stabilize the gradient or for a readout (cell fate decision) to be made at some finite time $t_f$ before the gradient vanishes. Such a readout could be rendered by, for example, the activation of a second set of regulatory elements for the relevant proteins. In the interest of defining a metric that is compatible with

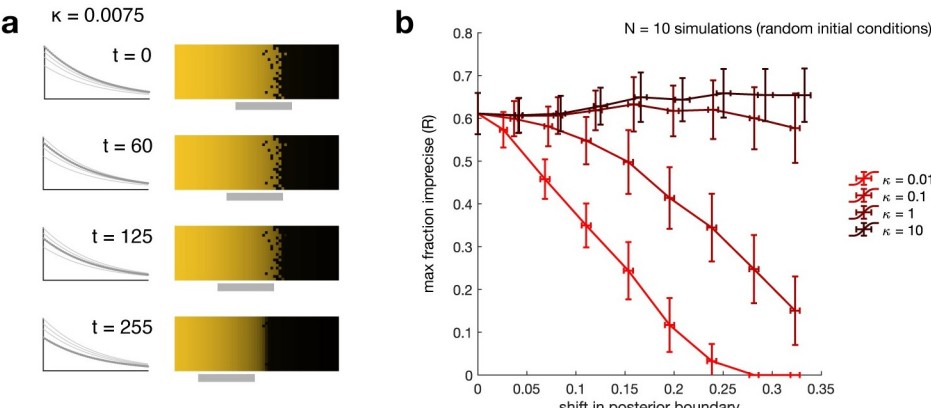

**Fig 3. (a) Temporal decrease in the morphogen concentration at every point in space can "sweep away" imprecision in the expression boundary**. Decrease in morphogen concentration causes the bistable region (solid gray line) to shift anteriorly over time. Thus, cells to the anterior are initially monostable for high $u_1^*$ and remain there as they become bistable, whereas cells in the initially bistable region become monostable for high $u_2^*$. The initial conditions here are derived from the steady-state solution to a system with a static gradient equivalent to that of the dynamic gradient at $t = 0$; see Fig 2 with $\kappa = \infty$. **(b) Slower gradient dynamics are more effective at increasing precision**. Lines are mean, error bars are standard deviation across $N = 10$ simulations with different initial conditions at different values of $\kappa$ (light to dark red), the ratio between rate of gradient decay and rate of convergence to steady state. $R$ quantifies the maximum width of the imprecise region relative to the width of the bistable region at the time $t_f$ by which the boundary has shifted by a fixed amount from its initial location. Lower $R$ corresponds to greater precision (see text). Slower-decaying gradients produce more precise boundaries for the same total shift than do faster-decaying gradients, but take longer to achieve these gains.

stabilization as well as independent of the readout mechanism (which might itself introduce error amplification or correction), we will assess the degree of precision enhancement as follows:

1. Let morphogen concentration decrease from time 0 to time $t_f$.

2. At time $t_f$, maintain the current morphogen profile and allow the network within each cell to approach steady state (let $t \to \infty$).

3. Calculate $R$ as in Eq (8), the fraction of the final bistable region which may be irregular.

Although $R$ is defined with respect to the width of the bistable region at infinite time (equivalently $t = t_f$), from it we can calculate the relative width of the irregular region to the width of the bistable region at any time $t$ for which it exists, as

$$R_t = \frac{x_p(t_f) - x_a(t_f)}{x_p(t) - x_a(t)} R, \tag{11}$$

where $x_p(t)$ ($x_a(t)$) is the posterior (anterior) boundary of the bistable region at time $t$. This is equivalent to comparing the width of the irregular region when the gradient is dynamic to a control with static gradient equal to the dynamic gradient at time $t$.

There is a tradeoff between the precision enhancement and the shift in the boundary, with greater boundary shifts resulting from longer times to readout. For a relatively slow gradient (small $\kappa$) there are clear improvements in precision (decrease in $R$), but high $\kappa$ does not generally reduce precision and may even exacerbate it to a small degree (Fig 3B). Combined with observations from Fig 2, these results suggest that precision enhancement mechanisms are most effective when network dynamics are more rapid than input dynamics. See also S1 Appendix for an illustration of this effect using a 2D autonomous model. We reiterate here

that the equivalence of $R$ to a metric of reproducibility when the initial conditions instead correspond to individual embryos shows that the precision enhancement mechanism could also reduce population variability in boundary placement.

We have chosen to focus this example on the case where the system transitions from monostable high $u_1^*$ to bistable to monostable low $u_1^*$. This is not a requirement, however, to carry out the same type of analysis for precision demonstrated here and in the preceding section. We also note that we can generalize this approach to other networks with time-varying bifurcation parameters other than morphogen concentration provided that (a) those parameters are independent of the concentrations in the explicitly modeled genetic network (i.e., all feedback loops must be included in the model, not as time-varying inputs), and (b) the temporal evolution of the bifurcation parameters is such that there is a monotonic progression from monostable high (low) to bistable and/or from bistable to monostable low (high), as in the case of the simple gradient explored above. Finally, while we have used random initial conditions in our example, the methodology and metric apply equally well when initial conditions are structured, and from a design standpoint may be employed to impose constraints on initial conditions if systems are to achieve a desired end result.

## Boundary shifts as propagating fronts

Diffusion of protein products is one way to average out stochastic variations between neighboring cells or nuclei, but it also permits classes of dynamic behaviors that are not observed in systems without diffusion. For example, it is well known that bistable systems with diffusion and spatially homogeneous parameters admit traveling wavefront solutions; biological examples include axonal propagation [45], population genetics [46], cell polarization [47], and Fgf8/RA gradient formation [48]. The speed and direction of front propagation depend on which of the two stable steady states has a larger basin of attraction.

Here, we propose that spatial variation in bifurcation parameters can be understood to modulate the local front velocity in bistable regimes, providing a quantitative approximation for how a front, once formed, will propagate along a domain. Local stability properties allow us to qualitatively predict where fronts emerge in response to arbitrary initial conditions. In this way, transient patterning behaviors can be analyzed apart from the behavior of individual cells, with diffusion providing the link between local and global dynamics. The approach is similar in spirit to previous work suggesting that expression domains can be predicted to expand or shrink based on terms arising from local field approximations [49], but the mathematical details and conceptual treatment differ.

In this section, we will consider the counterpart to Eq (1) with diffusion:

$$\frac{\partial u(t,x)}{\partial t} = f(x, u(t,x)) + D\frac{\partial^2 u(t,x)}{\partial x^2} \tag{12}$$

where $D$ contains the diffusivities of gene expression products. We will continue to refer to the local mono- or bistability of this system as the solutions at a particular coordinate $x$ to the corresponding system without diffusion, though local phase portraits do not predict individual cell trajectories when cell-to-cell coupling is present. At each coordinate we will also assign a local front velocity $c(x)$, which is the velocity of a traveling front solution to

$$\frac{\partial \tilde{u}(t,\xi)}{\partial t} = f(x, \tilde{u}(t,\xi)) + D\frac{\partial^2 \tilde{u}(t,\xi)}{\partial \xi^2} \tag{13}$$

where $x$ is fixed and $\xi$ is a dummy spatial coordinate. In other words, the local front velocity captures the speed and direction that a traveling front would propagate if the parameters were

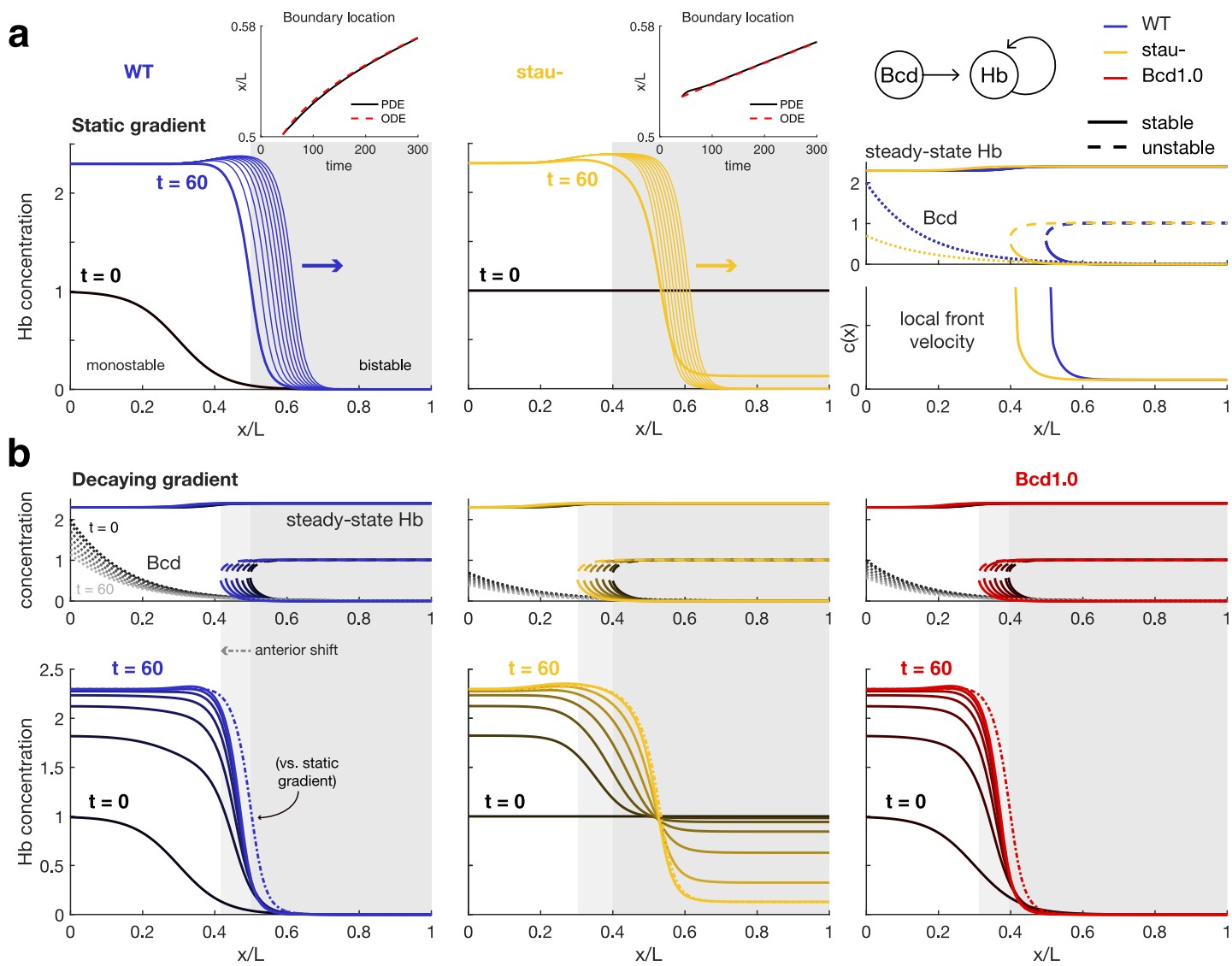

**Fig 4. (a) In the posterior of the embryo, the Hb boundary behaves as a bistable propagating wavefront**. Simulations (left, middle) and bifurcation diagrams (right) of the Hb patterning system with a static Bcd gradient show that the Hb gene expression boundary becomes a propagating bistable wavefront toward the posterior of the embryo (gray shaded region) and would continue to propagate slowly thereafter (thin lines) if not for external influences such as downstream genetic networks, cellularization, or gastrulation. Blue indicates wild-type (WT) and gold, *stau⁻* mutant. Right, the Bcd gradient for *stau⁻* (gold dotted line) has lower amplitude than that of the WT Bcd gradient (blue dotted line), which shifts the bifurcation (critical) point anteriorly. The local front velocity calculated from Eq (13) rapidly decreases posterior to the critical point but remains strictly positive, indicating that a bistable gene expression boundary will propagate posteriorly. The Bcd1.0 bifurcation diagram (see (b)) is almost identical to that of *stau⁻*. Top insets compare boundary location from PDE simulations (solid black) to the location from an ODE approximation (red dashed) derived from the local front velocities, with $\epsilon$ equal to a quarter of the boundary width. Simulation start $t = 0$ is 10 min before the beginning of nc14. **(b) Decay in the Bcd gradient shifts the bistable region anteriorly over time, causing boundaries to stabilize at slightly anterior positions**. Bottom, there is a greater discrepancy between boundaries forming over dynamic (solid lines) vs. static (dash-dot line) in the WT (blue) and Bcd1.0 (red) mutant than the *stau⁻* (gold) mutant due to differences in front emergence. Specifically, homogeneous initial conditions below the maximum value of the unstable steady state cause the boundary in the *stau⁻* mutant to appear within the bistable region, such that boundary propagation continues at roughly the same speed regardless of gradient dynamics. In contrast, in WT and Bcd1.0 the boundary is preinduced anterior to the critical point. A decrease in morphogen concentration in time causes the boundary to reach the critical point sooner, but also reduces the total time spent by the boundary in the region of rapid propagation, resulting in more anterior placement relative to the static case. All parameters are as given in [40].

spatially homogeneous at the values they assume at point *x* in Eq (12) (Fig 4A). Throughout this section, we will use the phrase *propagating front* to distinguish solutions to Eq (12) from traveling front solutions to Eq (13), which are translation invariant (i.e., their shape does not vary as they propagate); that is, for a propagating front, one or more of the bifurcation

parameters may vary in space along the domain, whereas for a traveling front these parameters are spatially homogeneous.

Given the local front velocities $c(x)$ at coordinates $x$, within a bistable regime it is possible to approximate the position of a propagating front $w(t)$ at time $t$ using the following ODE:

$$\frac{dw(t)}{dt} = c(w(t) + \epsilon), \qquad (14)$$

where $c$ depends on $f$ and $D$, and $\epsilon$ is a constant correction term that improves accuracy. This is to the best of our knowledge the first explicit such approximation applied in the biological literature. We compare the ODE approximation to full PDE simulation results for the two biological examples provided in this section; S2 Appendix illustrates an application to a 1D model where analytical solutions are available for $c$.

Conveniently for our applications, the front velocity can be calculated as a function of bifurcation parameters and then mapped to $x$ coordinates based on the spatial profile of those parameters. Thus, for each example in this section, we perform a one-time calculation of velocity as a function of morphogen concentration and use this relationship to deduce $c(x)$ for specific morphogen profiles. The front velocity at different diffusivities can be recovered by appropriately normalizing the $x$ coordinate. Increasing diffusivity increases propagation speed and also causes boundaries to become wider (S2 Appendix).

Analogously to the relationship between the time-independent dynamical system Eq (1) and the time-dependent one Eq (5), we can also add time dependence to Eq (12) by replacing $f(x, u)$ with some time-varying function $g(t, x, u)$. In this case, the local front velocity is also instantaneous to a fixed time $\tau$ (corresponding to solutions to Eq (13) with $f(x, \tilde{u}(t, \xi)) = g(\tau, x, \tilde{u}(t, \xi)))$.

The following subsection builds intuition about propagating wavefronts using an example from the developmental biology literature with one morphogen and one expressed gene. After introducing key concepts we apply our understanding of propagating fronts to a two-gene toggle switch that achieves scale invariant boundary placement with two opposing morphogen gradients.

**Example: Hb patterning over Bcd gradient in embryonic fruit flies.**   In a recent paper, Yang et al. quantified the anterior-to-posterior shift in the Hb boundary in wild-type, $stau^-$ mutant, and half-Bcd dosage (Bcd1.0) mutant *Drosophila melanogaster* embryos [40]. Here, we analyze their model to reveal that the Hb patterning system is split between a locally monostable anterior and a locally bistable posterior with an extremely low front velocity. Our purpose in this section is not to make new predictions about patterning in the living embryo, but rather to develop mathematical intuition behind propagating fronts in an explicitly biological context. In doing so we clarify the theoretical grounds for behavioral differences between the authors' models, and propose an alternative to temporal decreases in morphogen concentration to explain some of their experimental observations.

Hb patterning is modeled as a one-dimensional reaction-diffusion system with no-flux boundary conditions:

$$\frac{\partial h(t, x)}{\partial t} = f(b(t, x), h(t, x)) - \beta h(t, x) + D \frac{\partial^2 h(t, x)}{\partial x^2} \qquad (15)$$

where $h(t, x)$ and $b(t, x)$ are the concentrations of Hb and Bcd respectively at A-P axis coordinate $x$ and time $t$, $\beta$ is the decay rate of Hb, and $D$ is the diffusion constant of Hb. The

regulatory function

$$f(b,h) = \frac{\alpha_b \left( \frac{b}{b_0} \right)^{n_b} + \alpha_h \left( \frac{h}{h_0} \right)^{n_h}}{1 + \left( \frac{b}{b_0} \right)^{n_b} + \left( \frac{h}{h_0} \right)^{n_h}}$$

(16)

describes the activation of Hb expression by Bcd and Hb.

Consider the following dynamical system corresponding to Eq (15) without diffusion:

$$\frac{d\bar{h}(x)}{dt} = f(b(t,x), \bar{h}(t,x)) - \beta\bar{h}(t,x).$$

(17)

Further suppose that $b(t, x) = b(x)$ is fixed in time. Then the Bcd concentration $b(x)$ acts as a bifurcation parameter determining the local mono- or bistability of the corresponding reaction-diffusion system Eq (15). Specifically, with all other parameters as given in [40], then with $b(x) = 0$ the system Eq (17) is bistable, while for $b(x) > b_{crit} = 0.07$ the system is instead monostable for high Hb. An earlier model of Hb patterning also implicated Bcd in modulating bistability across the length of the embryo [50].

Returning to the full model (15), the Bcd gradient is described by

$$b(t,x) = b_m e^{-\frac{x}{\lambda} - \omega_0 (\max(t,t_0) - t_0)}$$

(18)

where $b_m$ controls the amplitude of the gradient, the length constant $\lambda$ determines the dropoff, and $w_0$ is the rate of decrease in Bcd concentration starting at time $t_0$. At all times $t$ during nc14, a transition from locally monostable toward the anterior of the embryo and locally bistable toward the posterior occurs at the coordinate $x_{crit}$ where $b(t, x_{crit}) = b_{crit}$. The bifurcation diagram reveals that the values of the high and low stable states in the bistable region are relatively constant some short distance from $x_{crit}$ owing to the low absolute change in $b(x)$ as the gradient decays in space (Fig 4A).

First, we use our knowledge of local dynamical properties to examine how initial conditions develop into propagating gene expression boundaries. Specifically, chemical concentrations at a coordinate $x$ will approach whichever steady state has a basin of attraction containing the initial condition at $x$. Hence, boundaries form at transitions from monostable to bistable regions, or where initial conditions cross from one basin of attraction to another. In WT and Bcd1.0, the initial condition is itself a boundary (with reduced magnitude) in the locally monostable anterior of the embryo. This initial boundary decays toward the value of the low stable state within the bistable region, such that in the absence of diffusion, the steady-state Hb in this region would remain at 0. A propagating front is initiated by the diffusion of Hb produced in the monostable anterior into the bistable posterior. Halving Bcd concentration to produce Bcd1.0 shifts $x_{crit}$ to the anterior, thereby hastening the approach of the front to the locally bistable region where propagation is slow.

In contrast, in $stau^-$ mutants the initial condition is homogeneous in space and falls just within the basin of attraction of the low stable state in the locally bistable posterior. Thus the boundary forms by the simultaneous diffusive influx of Hb from the monostable anterior and the attraction of Hb to the low state in the bistable posterior. Since the boundary evolves simultaneously toward both the high anterior and low posterior states (instead of beginning already at the low state in the posterior), the front is initially less steep than boundaries for the other two models. The large shift in Hb boundary position reported by the authors of [40] likely arises as a consequence of the horizontal compression of the boundary from an initially large and not very steep width to the sharp bistable front (Fig 4B).

Assume for now that the Bcd gradient is static. In the bistable region, we can analyze the local front velocity as defined in Eq (13). We find in all three models that it is essentially constant in the posterior of the embryo (Fig 4A). The propagation speed is slow but nonzero, such that although the boundary is theoretically transient, each minute of delay in the readout would incur an additional error of less than 1 μm in boundary placement (less than the average distance between nuclei). Since the front speed is quite large near $x_{crit}$, however, much faster propagation could theoretically be attained by a small constant offset in gradient concentration or by flattening the gradient at a low value. In Fig 4A we also compare the boundary location, defined as the point at which the boundary crosses the local value of the unstable steady state, in the full simulated PDE and in the ODE approximation beginning at $t = 42$ min, or 32 min into nc14 (when the front first forms, i.e., first crosses the unstable steady state). The inital bump in boundary location for the *stau*⁻ PDE simulation results from the fact that the posterior side of the boundary has not yet stabilized at 0 concentration as it will in the profile for the final propagating front. Even with this deviation, the error in the ODE approximation for all three models (Bcd1.0 not pictured) is at most 0.2% of embryo length, or about 2.2% of boundary width, for a simulated time span up to $t = 600$ min (well past the time at which the embryo has entered gastrulation).

The effect of temporal decrease in Bcd concentration is to shift $x_{crit}$ anteriorly over time, which hastens the "capture" of emerging fronts in the locally bistable region of slow propagation, causing the fronts to occupy a slightly anterior position relative to predictions with static gradients for WT and Bcd1.0. Given the uniformity of the local front velocity in the posterior, we predict that temporal decrease in Bcd once a front is already in the bistable region should have little effect on boundary location. Indeed, since the front in the *stau*⁻ model emerges within the bistable region for both static and dynamic gradients, we see little difference between simulations with and without temporal decrease in Bcd (Fig 4B).

Viewing the Hb boundary as a propagating wavefront suggests a straightforward mechanism for stabilizing the pattern—namely, to reduce protein diffusivity to 0. Experimental measurements indicate that effective diffusivity, or internuclear coupling strength, begins to decrease 20 min into nc14, reaching 0 by 30 min [20]. This time window coincides with the time the boundary reaches the bistable region in the WT embryo in our simulations. It is worth noting that 20 min also marks the start of a reduction in slope in the data for *stau*⁻ reported in Fig 6 of [40], while 30 min is the point at which their model predictions for Bcd1.0 boundary placement also cease to recapitulate experimental results. In the WT simulations, the Hb boundary at 20 or 30 min for a static gradient is reasonably close to the location of the boundary for the system with a dynamic gradient at 20, 30, and 50 min into nc14 (S1 Fig). Temporal decrease in morphogen concentration after the loss of diffusion would not be expected to change boundary position, since cells to the anterior of the boundary would maintain their high state as the underlying region became locally bistable. It is therefore worth considering to what extent the gradient dynamics are in fact transmitting new dynamic positional information, as opposed to being primarily symptomatic of a full handoff in control of the Hb boundary to downstream networks [51].

Finally, in the example shown here, boundary placement for propagating wavefronts behaves rather similarly to a traditional thresholding model with respect to embryonic scaling; that is, in the absence of other adjustments, changing embryo length without altering the absolute scale of the gradient should negligibly affect the absolute position of the boundary and therefore produce an anterior or posterior shift with respect to normalized coordinates, depending upon whether the embryo were lengthened or shortened. In practice, however, gradients may change shape with embryo geometry. For example, experiments conducted in *D. melanogaster* with artificially shortened embryos produced Bcd gradients later and lower in

amplitude than WT, resulting in a posterior shift in the Hb boundary of just under 4% percent embryo length [52], as compared to the approximately 12% that would be otherwise predicted for embryos 80% the length of WT. Even if gradients were to scale perfectly, our simulations indicate modest effects on boundary placement when initial conditions are not also scaled. With the dynamic gradient, the additional shift is less than 0.5% for embryos at 80% WT length, but rises to almost 2% for embryos 50% WT length (S1 Fig), at which point cells within the bistable region begin with nonzero Hb concentration. Further research would be necessary to determine whether very large or small embryos suffer from robustness issues due to compounding errors, as well as whether initial conditions of fixed length scale have evolved to be compatible with a range of scales downstream.

**The final front(ier): Accurate boundary placement through front localization.** Thinking about gene expression boundaries as propagating fronts reveals mathematical designs for pattern-forming systems yet to be found in nature or constructed synthetically. Here, we consider the case where the local front velocity passes through zero at some critical point in the domain. This implies a switch in propagation direction, such that a front forming on either side of the point will propagate toward it. This scheme decouples the steady-state location of the boundary from the exact value of the homogeneous initial condition, and also produces qualitatively predictable transient behaviors. Once a localized boundary is established, its stability to intrinsic and extrinsic perturbation may be further analyzed to assess its reliability as a cue for downstream patterning processes [38, 53]. This scheme is distinct from recent research into wave pinning, whereby two mutually repressive species with highly disparate diffusivities may also produce bistable fronts that are fixed in place. In these models a highly diffusive species acts as the bifurcation parameter, and its decay over time "pins" a propagating boundary in place [47, 54, 55]. In other words, propagation is halted through the temporal modulation of a homogeneous-in-space parameter controlling the front velocity. In the model discussed in this section, propagation is stopped rather through spatial modulation of exogenous parameters that locally bring the front velocity to zero.

A number of developmental patterning systems employ opposing morphogen gradients, perhaps to enhance accuracy in boundary placement [56] or confer robustness to tissue scaling [9, 57]. We employ a simple model of a chemical toggle switch in which two mutually repressive species $u_1$, $u_2$ are activated by morphogens $\alpha_1$, $\alpha_2$ respectively. The morphogens are expressed in opposing gradients. For simplicity, transcription and translation are lumped into a single step. The governing equations are

$$\begin{cases} \frac{\partial u_1(t,x)}{\partial t} = a h_a(\alpha_1(x)) h_r(u_2(t,x)) - \beta u_1(t,x) + D_1 \frac{\partial^2 u_1(t,x)}{\partial x^2} \\ \frac{\partial u_2(t,x)}{\partial t} = a h_a(\alpha_2(x)) h_r(u_1(t,x)) - \beta u_2(t,x) + D_2 \frac{\partial^2 u_2(t,x)}{\partial x^2} \end{cases} \tag{19}$$

where $h_a$, $h_r$ are activating and repressing Hill functions respectively, as in Eq (4).

Assume $D_1 = D_2$. The front localizes at the point $x_l$ where the switch is balanced such that neither of the two stable steady states is dominant, i.e., $\alpha_1(x_l) = \alpha_2(x_l)$ (assuming $x_l$ is sufficiently far from the edges of the domain). Therefore, in this scheme, the final placement of the gene expression boundary relative to the domain edges will be unaffected by changes to $\alpha_1(x)$ and $\alpha_2(x)$ that do not shift $x_l$, e.g., scaling the embryo length or equally modulating the amplitude or spatial decay rates of the gradients (provided that bistability is maintained in the region near $x_l$). Quantitative features such as the boundary steepness and propagation speed may still depend on the exact parameter values (S2 Appendix).

From a bifurcation plot of the local steady states, we can predict that uniform-in-space initial conditions will produce a front. Notice that here we have collapsed the effect of two

bifurcation parameters, $\alpha_1(x)$ and $\alpha_2(x)$, onto a single bifurcation parameter, the axis $x$. The switch in the anterior of the domain is biased in favor of $u_1$ while the posterior is biased in favor of $u_2$, such that there is a range of concentrations for which initial conditions homogeneous or near-homogeneous in both chemicals will fall into the basin of attraction of $u_1$ toward the anterior and $u_2$ toward the posterior. This means that a properly oriented front (high $u_1$ in the anterior) will emerge and approach the localization point.

The initial location where the front appears depends on a combination of the numerical values of the initial concentrations, and the relative rates of front emergence (local divergence toward high or low steady state) vs. local front speed at the point where the initial concentrations cross from one basin of attraction to the other. Although the final boundary location may be fixed, the transients vary depending on whether $u_1$ or $u_2$ is favored by the initial conditions, since the boundary may approach its final position from left or right. If these transients are important for downstream patterning networks, the initial conditions should be constrained to have the appropriate asymmetry. Following front formation the behavior will then be relatively robust to the exact degree of asymmetry. In contrast, a boundary forming in a system without diffusion would remain at the initial location where it appeared, rendering the final placement sensitive to initial conditions (Fig 5A). The transient, however, would follow roughly the same dynamics on either side of the boundary, without any dramatic shift. Since propagation speed decreases with decreasing diffusivity, we can consider the case with no diffusion as the limiting case of a diffusive system for low $D_1$, $D_2$. From this perspective, we see that diffusion provides a means to balance the tradeoff between accuracy in boundary placement, the degree of accuracy required of the initial conditions, and the dependence of downstream components on transient behaviors upstream.

If $D_1$ and $D_2$ are unequal, then the localization point no longer corresponds to the point where the switches are locally balanced. Specifically, the localization point $\tilde{x}_l$ shifts toward lower concentrations of the morphogen that activates the faster-diffusing species. Thus, differences in diffusivity reverse the direction of front propagation in the region between $x_l$ and $\tilde{x}_l$. Larger disparities increase the size of the shift (Fig 5B).

Future quantitative investigation of developmental biological systems may reveal whether the front localization mechanism occurs in natural contexts. In the meanwhile, an understanding of propagating fronts can inform attempts to engineer synthetic multicellular patterning. For example, researchers have recently implemented patterning over concentration gradients with bistable networks in bacteria [58, 59]. Our results suggest that adding diffusion of gene expression products to such systems could enable dynamic, predictable boundary placement without the need to manipulate bifurcation parameters in real time. We may also envision more complicated patterning schemes for natural or synthetic application, particularly in systems where a single domain possesses multiple localization points (Fig 6).

## Discussion

Thresholding models for morphogen gradient interpretation are susceptible to variation in initial chemical concentrations and the shape of morphogen gradients [60] as well as to intrinsic stochasticity in gene expression [29, 38, 61, 62]. Dynamic positional information has been proposed as a framework to discuss the gradual refinement of gene expression boundaries [62] and to better recapitulate empirical observations [17]. In this work, we have argued that the means by which bistable gene expression boundaries refine over time is distinct between systems in which gene products diffuse and those in which they do not. We have illustrated that a deterministic model without diffusion isolates the dynamics of neighboring cells, such that temporal evolution in morphogen concentrations can drive the boundary toward transitions

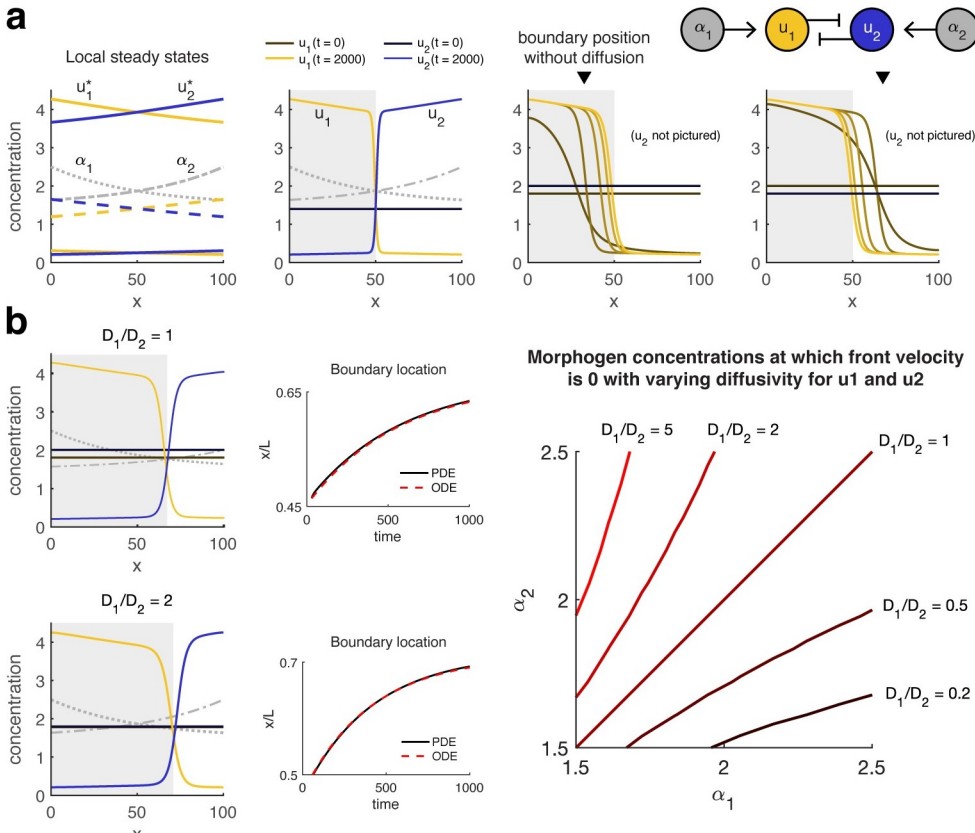

**Fig 5. (a) Diffusion permits a bistable gene expression boundary to be accurately placed independently of initial conditions**. Left, bifurcation diagram and right, simulation results for Eq (19) with $a = 1.7$, $\beta = 0.35$, $D_1 = D_2 = 1$, $n_a = n_r = 2$, $K_a = 0.75$, $K_r = 1$, and $\alpha_1(x)$, $\alpha_2(x)$ as shown. Gray shaded region indicates rightward propagation of the front in $u_1$ (gold) and leftward for the front in $u_2$ (blue), with lighter shades indicating later time points. Regardless of the initial condition, the boundary propagates toward the point where the toggle switch is "balanced" (the local steady states are such that $u_1^* = u_2^*$). Inverted black triangle indicates the position of a gene expression boundary with the same initial conditions when gene products do not diffuse. For clarity, only $u_1$ is pictured in the third and fourth plots. **(b) Imbalance in gradients or diffusivities may shift the location of the boundary**. Right, when $D_1 = D_2$ (top), the boundary localizes where $\alpha_1(x) = \alpha_2(x)$. Letting $D_1 = 2 = 2D_2$ (bottom) shifts the localization point in favor of higher $u_1$. Middle, ODE approximations from the time of boundary formation produce an error less than 0.5% embryo width in predicting boundary location. Simulations are shown for $\epsilon = 0$ ($D_1 = D_2$) or $\epsilon = -0.5$ ($D_1 = 2D_2$). Right, changing the diffusion ratio shifts the nullcline for front velocity toward lower concentrations of the faster-diffusing species. Decreasing the diffusivity also decreases the width (increases the steepness) of the respective front (S2 Appendix).

from bistable to monostable regions in the embryo. Allowing gene products to diffuse removes the direct link from morphogen concentration to individual cell dynamics, but local morphogen concentrations can still be used to quantitatively predict boundary shifting across bistable regions toward points in the reaction domain where both steady states are equally dominant.

We suggest there are design tradeoffs among the accuracy of boundary placement, boundary precision, the time to readout, and the amount of inaccuracy tolerated in the initial conditions. Cell-to-cell variation in initial conditions contributes to boundary imprecision, while embryo-to-embryo variation in initial conditions—even if initial conditions are uniform within a given embryo—can reduce reproducibility in long-term boundary placement in systems where protein products do not diffuse. Time evolution in morphogen gradients can alleviate the unwanted effects of inaccurate or imprecise initial conditions provided that the rate of evolution is slow with respect to network dynamics, though slower gradients also require

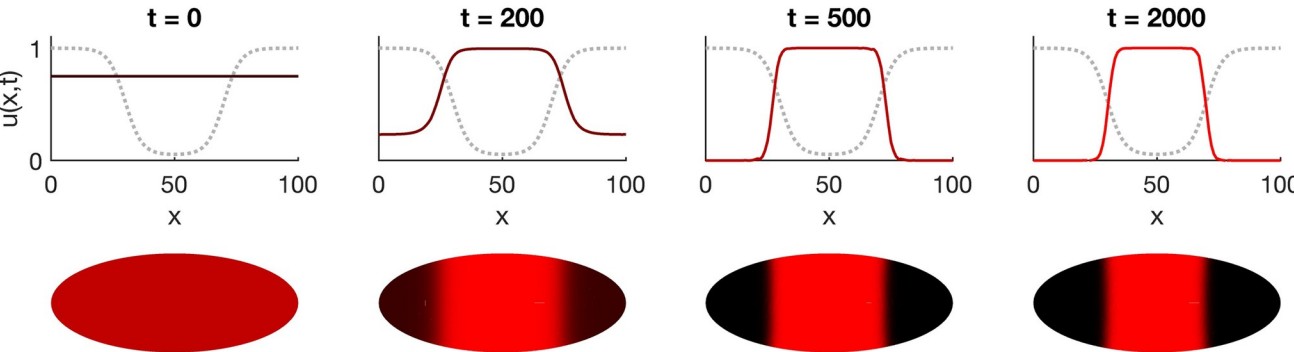

**Fig 6. Fronts may localize at multiple points in a domain, resulting in more complicated patterns.** Simulations were performed on the 1D model in S2 Appendix with $a = 0.1$, $D = 0.1$ on a 2D reaction domain (bottom) in which the bifurcation parameter $\alpha(x)$ varies only along the horizontal axis (gray dashed line). Top, horizontal cross-sections through the midline show that two opposing fronts appear and propagate toward the nearest localization point, resulting in refinement of a stripe pattern from either direction over time.

longer times to readout. If protein products are permitted to diffuse, random variations in initial conditions among cells are expected to average out, but boundary placement will not generally remain constant in time. However, certain combinations of network parameters and morphogen profiles can localize boundaries at a fixed point within a domain. In general, if downstream networks depend on the transient behaviors of propagating boundaries, then readout times or initial conditions will become more constrained. Finally, we note that diffusivity may also vary in time, for example, during cellularization in *Drosophila* embryogenesis. It is worth considering whether such a transition to more restricted cell-to-cell interaction acts as a compromise between washing out initial variation in concentration and constraining continued boundary movement.

A common challenge in the field of biology is that multiple models may produce substantially similar behaviors. We noted that the experimental results from [40] concerning the placement of the Hb boundary are both well described by a model with a decaying Bcd gradient and a model in which diffusion decreases over time. Similarly, on a qualitative level the gradual refinement in precision of a gene expression boundary can be described both by the dynamic morphogen model proposed here and by a model with a static gradient and stochastic system evolution [29]. Determining whether a model accurately captures an empirical behavior requires systematic and often quantitative comparison of theoretical predictions with experimental results. For example, the mathematical effect observed in [29] to refine gene expression boundaries in stochastic models was empirically verified in vertebrate neural tube through a careful interplay of theory and experiment [27]. We encourage the community to follow the example of these authors in future efforts to validate a proposed model, or to differentiate among potentially many mathematical descriptions of the same phenomenon.

In stochastic models such as the aforementioned, gene expression noise shapes the precision and positioning of a boundary by enabling stochastic switching, in which a trajectory randomly crosses from one basin of attraction to another [28, 29]. A logical extension to these works and to ours would be to examine boundary formation in stochastic networks with time-varying morphogen gradients. Though our simulations indicate that imprecision can be completely erased under certain circumstances, in a stochastic system some level of variation will persist regardless of how slowly the gradient evolves. We expect the ratio of variation in initial conditions vs. gene expression noise would affect the emergence or erasure of imprecision over time. Further experimental work would also be necessary to elucidate the extent to

which initial conditions do in fact vary in natural patterning systems, and what the causes of that variation or relative lack thereof might be.

In its current form, the ODE approximation to the dynamics of boundary position owes its justification to an engineer's reckoning rather than formal mathematical derivation. One of the limitations, therefore, concerns the adjustment parameter $\epsilon$. For systems with equal diffusivity among species and for which the domain is everywhere bistable, $\epsilon = 0$ appears to produce very good estimates (Fig 5 and S2 Appendix). Otherwise, the highest accuracy of prediction occurs for $\epsilon$ nonzero. We hypothesize that $\epsilon$ might theoretically be derived from boundary shape. In particular, the ODE approximation assumes a single velocity for a front, but in a spatially nonhomogeneous system a front will straddle multiple local front velocities. The parameter $\epsilon$ selects what point along the boundary is taken to approximate the velocity. In the event of a transition from a monostable to bistable region, for some period of time the boundary will straddle the bifurcation, and therefore the leading edge of the boundary may produce a better estimate of front velocity than the crossover with an unstable steady state. Similarly, once diffusivity is not the same for two species, the width of the boundary will vary by species, so again the speed may no longer be best approximated at the crossover point. Other sources of error could arise from the fact that a true traveling wave, from which the local front velocity is derived, may have slightly varying boundary shape depending on the velocity (S2 Appendix), and will propagate in an infinite domain, unlike the finite-length embryos considered here. A formal mathematical investigation of the approximation could help to guide the selection of $\epsilon$ (or alternative adjustment parameters) and place theoretical bounds on the expected prediction error. Further investigation would also help to develop metrics to assess where fronts emerge from given sets of initial conditions, and to improve the accuracy of the ODE approximation Eq (14) for time-varying $c(t, x)$ in the vicinity of bifurcation points.

Although we have taken a bistable switch as our conceptual model, we believe the basic ideas explored herein may be generalized to a broader class of systems. For example, our hypothesized mechanism for enhancing boundary precision essentially relies upon the ability of cells to be preinduced to a desired state before temporal changes introduce further bifurcations. Such (evasion of) topological capture as a means to correct for variation in initial conditions could generalize to more complicated multistable networks. In such cases it may be appropriate to draw upon mathematical tools from the growing literature on nonautonomous dynamical systems; for example, the notions of pullback attractors and tipping points could be employed to determine the maximum rate at which bifurcation parameters can evolve in time without disrupting the ability of a trajectory to track a desired steady state [63, 64].

Similarly, propagating fronts could be found in bistable systems of higher than two dimensions with many more bifurcation parameters than considered here. With appropriate care, bistable subsystems might even be dissected from more complex networks, with the contributions of those networks serving as time-varying parameters or as factors that place bounds on dynamic behavior. Traveling wavefronts in non-bistable systems are also known to exist, for example, in monostable systems, although unlike bistable fronts there is a minimum speed at which they must propagate [65]. Continuing mathematical investigation will doubtless open up new avenues for exploring propagating fronts in gene expression patterning and other areas of embryonic development [66].

Even working from a simple bistable model, our analysis highlights a few ways in which dynamic positional information may be further explored to better our understanding of gene expression patterning, both philosophically and mathematically. Consider signaling duration, which has emerged as a central feature of morphogen-signaling systems [14]. From a dynamical systems perspective we might expect signal duration to matter even with a static gradient: Supposing a signal modulates local mono- or bistability, then if a cell begins from a low initial

condition, it may take longer to approach a high expression state than a low one simply because it takes more time to produce more protein than to produce less. This observation may partially explain some experimental results, e.g., that anterior patterning in embryonic *Drosophila* is more highly disrupted by transient perturbations to areas with high than low Bcd concentration [13]. Signaling duration also plays a role in our two hypothetical mechanisms. In the bistable sweep model, cells to one end of the boundary require a minimum amount of time to "lock in" to the appropriate steady state before a bifurcation arises elsewhere in phase space. As we found in our examination of propagating fronts, signaling duration in one region of a tissue can also determine fates in other regions of the tissue simply because a gene expression boundary requires time to propagate. In the case of Hb patterning, decreasing signal in the anterior of the embryo decreased propagation speed, thereby shifting the entire boundary location anteriorly.

In addition to explicitly incorporating temporal evolution, dynamic positional information has implicitly challenged the notion of threshold-dependent activation, which we argue conflates the influence of two factors: the morphogen and the initial condition. In the simple boundary-forming model in Fig 1A, the genetic network does indeed act as a threshold detector for very high or low morphogen concentrations, since cells in these regions are monostable and therefore their final fates are independent of initial conditions. However, there is a band of concentrations for which the system is bistable and cells must be in a responsive state induced through initial conditions or coupling with neighbors in order to approach the appropriate final expression level. Thus, for a fixed uniform initial condition, the apparent threshold at which cells are activated will shift, giving rise to the observed inaccuracy in boundary placement between naïve thresholding (French flag) models and empirical observations.

Local threshold-dependent activation also cannot explain boundary placement in the bistable sweep or front localization designs, even though both exhibit predictable behavior regardless of initial condition. In the bistable sweep as illustrated in Fig 3, the sharp boundary is ultimately found at the posterior transition from bistable to monostable, which shifts anteriorly over time. In the front localization design, the boundary position is fixed across the whole range of initial conditions that produce a boundary, such that a "threshold" for activation is determined by the single point where the front velocity crosses zero. In both cases, individual cellular trajectories have less predictive power for the final observed pattern than do global dynamical properties.

The matter of thresholding and initial conditions raises a further philosophical point in relation to the notion of cellular competence, or the ability to "[express] all the necessary components for receiving a signal" [14]. Consider a bistable autoregulatory element in which species $A$ activates its own expression and is also activated by a signal $S$ that acts as a bifurcation parameter for the switch. Further suppose an upstream network component must establish nonzero $A_0$ as an initial condition; otherwise $A_0$ defaults to 0. Under these circumstances, the concentration of signal $S$ can determine whether $A$ approaches a high state or decays to a low state, based on which basin of attraction contains $A_0$. However, if the system lacks the upstream network component, then $A_0 = 0$ and consequently $A$ will approach the low state regardless of the value of $S$. This would seem to indicate that $A$ is not responding to the signal $S$, but this situation is clearly distinct from the loss of the autoregulatory interaction of $A$ upon itself or the loss of the capacity for $S$ to promote expression of $A$, both of which would destroy the ability of $A$ to respond to $S$ in switchlike fashion regardless of initial condition. Of course, it is rare that real developmental systems decouple nicely into components that only establishing initial conditions vs. those that govern subsequent dynamic evolution. Nevertheless, the distinctions made clear through a precise mathematical representation suggest potential areas to refine our qualitative understanding.

Taken together, our observations support the notion that positional information is neither locally contained nor statically interpreted. Dynamic features may play functional roles in enhancing patterning robustness to initial conditions. At the same time, tissue-level patterning behaviors may not be immediately predictable from examining individual cell trajectories, especially in the presence of cell-to-cell coupling. Future research in gene expression patterning will benefit from further techniques for modeling temporal evolution at multicellular scales, particularly with respect to transient dynamic behaviors. Whether the hypothetical mechanisms explored here are employed in natural embryonic development remains to be seen. Nevertheless, we are confident that their presence or absence will prove insightful in our quest to disentangle explanation from observation, function from form.

## Methods

All simulations were performed in Matlab 2017b. Code is available in S1 Code.

## Supporting information

**S1 Code. Matlab code for all simulations referenced in the text.**
(ZIP)

**S1 Fig. Additional plots for Hb boundary placement with and without dynamic gradients and scaling.**
(TIF)

**S1 Appendix. 2D autonomous model for precision enhancement and expressions for convergence rate to steady state.**
(PDF)

**S2 Appendix. 1D model for front localization on a finite domain.**
(PDF)

## Acknowledgments

The author would like to thank Murat Arcak for suggesting a numerical method to verify stability of 1D localized front solutions in finite domains. The author also thanks Justin Yim for providing code to approximate the separatrix of a 2D bistable system, and Regina Eckert for providing code to generate tile plots.

## Author Contributions

**Conceptualization:** Melinda Liu Perkins.

**Formal analysis:** Melinda Liu Perkins.

**Funding acquisition:** Melinda Liu Perkins.

**Investigation:** Melinda Liu Perkins.

**Methodology:** Melinda Liu Perkins.

**Resources:** Melinda Liu Perkins.

**Software:** Melinda Liu Perkins.

**Validation:** Melinda Liu Perkins.

**Visualization:** Melinda Liu Perkins.

**Writing – original draft:** Melinda Liu Perkins.

**Writing – review & editing:** Melinda Liu Perkins.

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
