## [Decision Letter · Decision Letter 0]

8 Jan 2021

Dear Dr. Perkins,

Thank you very much for submitting your manuscript "Dynamic positioning and precision of bistable gene expression boundaries through diffusion and morphogen decay" for consideration at PLOS Computational Biology.

As with all papers reviewed by the journal, your manuscript was reviewed by members of the editorial board and by several independent reviewers. In light of the reviews (below this email), we would like to invite the resubmission of a significantly-revised version that takes into account the reviewers' comments.

We cannot make any decision about publication until we have seen the revised manuscript and your response to the reviewers' comments. Your revised manuscript is also likely to be sent to reviewers for further evaluation.

Sincerely,

Attila Csikász-Nagy

Associate Editor

PLOS Computational Biology

Florian Markowetz

Deputy Editor

PLOS Computational Biology

Reviewer's Responses to Questions

**Comments to the Authors:**

Reviewer #1: In the manuscript the author proposes different dynamical mechanisms that can affect the position of gene expression spatial patterns during embryo development. The mathematical and computational aspects are correct, nevertheless I found the manuscript is unfocused and rushed. While any of the two mechanisms studied - gradient decay and TF diffusion - are interesting, they are not explored in depth enough for a scientific publication. As a personal opinion, I would encourage the author to separate both topics in different papers and explore them properly trying to apply them to different scenarios. From the current version of the paper it is clear that the author has the tools and understanding of the problem to create a higher impact paper. Here are some other comments on the manuscript:

- In the section "Enhanced boundary precision through temporal gradient decay" the author proposes that initial condition variability is reduced by the gradient decay. In order to check the validity of this, the author should show a comparison of the pattern formation with and without gradient decay to explore the differences.

- The author also uses an arbitrary initial condition in which each cell shows a random gene expression. It is not clear to me that this is biologically relevant. In the current scenario cells are kept in a noisy naive state despite being already in the presence of the gradient. A more realistic scenario would include the fact that the cells are already in that gradient by either assuming that each cell is in equilibrium with the gradient before its decay start or by including gradient formation in the formulation of the problem.

- I found interesting the analysis of the gradient decay from a bifurcation point of view. Nevertheless in general the same effect would be true for any other varying parameter of the gene regulatory network. If the author wants to focus exclusively on the the decay of the gradient, a more detailed analysis of the biological timescales is required e.g. is the decay of the morphogen required to be slower than the decay of the GRN transcription factors? What happens with the boundary when this timescale separation does not hold?

- In the same section, one of the limitations of the mechanism proposed is that once the gradient decays completely, the boundary will completely disappear (there will only be a monostable u_2 regime). What are the time limit constrains for cell fate decision in this aspect?

- In the second part of the manuscript "Boundary shifts induced by near-traveling fronts". It is not clear to what is the benefit of the mathematical formulation. I expected to see some applications of formula (9) in the manuscript. If I understood correctly, all the analysis is done by performing numeric integrations of formula (8), and in particular the same exact formulas than the ones presented in [37].

- The application fo the Bicoid-Hunchback system was interesting. The author dissected the role that diffusion and gradient decay have in this particular system. Nevertheless, in its current state is just an analysis on a toy-model formulated in a different paper and it was not clear the biological impact of the results. For instance the author analyses what happens at times greater than 60 minutes in the static gradient case (thin lines in Fig 2a,b) but it does not determine what timepoints were used for these lines or why it is relevant in the fly patterning that the boundary stops moving. In particular we know that boundaries in the fly move while they pattern as a result of the GRN dynamics of the GAP genes.

- Similarly, it looks that the initial condition is very important in the positioning of the boundary. Nevertheless I am not sure to which extent the proposed initial conditions are an artefact used in the toy model proposed in [37]. In this aspect I agree that the author has just replicated the same conditions, but then I missed further exploration of this property.

- On the same note, in the last section of the paper (Fig. 3). The author requires that the initial levels of the u1,u2 are intermediate levels of expression. Similarly to the initial questions, does this imply that there was no cross-repression until the start of the simulation? What happens when the levels are already in equilibrium with the gradient? What happens if they are not in equilibrium of the gradient but then they follow gradient formation?

- lines 400-403 "Thus, for a fixed uniform initial condition, the apparent threshold at which cells are activated will shift, giving rise to the observed inaccuracy in boundary placement in such models.". I do not think that the word "inaccuracy" is accurate. We know that boundaries move in many embryonic systems. The idealisation that boundaries must be static is an artifact of our imposition of the French flag model.

- Whilst it looked interesting, I could not fully follow the argument of paragraph 414-428 in the discussion.

Some other minor comments:

- line 105 is not clear what the author means by "cell behaviour can be local", or "cell behaviour can be instantaneous".

- line 146 "The speed and direction of front propagation depend on parameter values, specifically on which of the two stable steady states is more favourable". It is not clear what a "more favourable" state means or how it is related to the parameter values mentioned in the first part of the sentence

- line 196 "ambryo" should be "embryo"

- lines 163-169 the definition of 'near-travelling' front could be a bit clearer

Reviewer #2: Please see attached.

Reviewer #3: Melinda Liu Perkins constructs general bistable genetic networks driven by a morphogen gradient, and investigates the influence of the two factors: dynamic decay of morphogen gradients and diffusion, on precision and shifts of the gene expression boundaries. Two representative examples show that these two factors could improve the precision of the boundaries. This work could deep our understand on the role of dynamic positional information on the noise filtering mechanism in morphogen-based patterning. However, the author needs to address the following questions before publication.

1. The author needs to show more details on the role of the morphogen decay in the refinement of the boundary in the toggle-switch model shown in Figure 1. The repression of u2 on u1 could also help to refine the boundary as the concentration of u2 increases. Hence it is necessary to show the boundary evolution without morphogen decay as a control. It also would be better that the author could run some parameter sensitivity analysis to show the robustness of the conclusion, e.g., vary the decay rate of the morphogen.

2. It is interesting to see the Hb boundary keeps shifting after 60 min into nuclear cycle (nc) 14 in the simplified model shown in Figure 2. But the author needs to point out that such a phenomenon is not biological relevant. As it is well known that the Hb expression is driven by the Bcd activation and self-activation (P2 enhancer) only before early nc14. Its regulation is switched to the stripe enhancer without Bcd activation and self-activation but repression from other gap genes such as Kni and Kr.

3. it is also necessary to do the parameter sensitivity analysis (especially on the diffusion constant) on the role of diffusion in pattern refinement shown in Figure 3. In the gap gene network, it is often believed that the diffusion role is very limited. So it would be better that the author could discuss some biological relevance.

4.Temporal refinement in gene expression domains has been investigated with the gene circuit model of gap genes, e.g., Manu et al., PLos Bio, 2009. So the first sentence in the Abstract is not accurate.

Reviewer #4: Disclaimer: even if I am quite familiar with mathematics, I do not have the background knowledge ready nor the time to look carefully at all the equations and verify/confirm they are correct. As a consequence, this review focuses on the interface biology-maths. I would like to suggest the editor to complement my review with one by a person better trained in the relevant maths.

Having said that, in this manuscript, the author develops and applies mathematical theory to further our understanding of the framework of dynamical positional information and its consequences for developmental patterning. As explained by the author, dynamical positional information was formulated partly in response to experimental findings that show (1) plenty of variation in initial conditions (messy gene expression patterns) is resolved into refined, clean final patterns, and (2) expression domains tend to shift over time across the tissue. The author then proceeds to study two popular explanations, morphogen decay and protein diffusion, in the context of boundary precision and boundary shifting. She uses the famous toggle switch network to illustrate her mathematical insights.

These insights are: (1) morphogen decay causes the bistable regions to shift, which forces cells that were initially in the bistable region (where one has least precision) into a monostable one. The result is a sharpe, and shifted, boundary. And (2) diffusion can generate boundary shifts, which may be halted by a gradient. In contrast to the first insight, here the shift may be in both directions. In both cases, the author then proceeds to establish certain requirements that need to be fullfilled for these mechanisms.

I find the manuscript a pleasure to read. It is clearly written and the figures are in very good shape. Of course, I do have a few comments for the author:

Major comments:

I would emphasize the differences between the two ideas in terms of what they can do. I am mainly thinking about how the first has "the boundary shifting up the gradient" and the second tends to shift "down the gradient". It would make the article stronger if the author presents two orthogonal ideas, instead of 2 variants on a single idea. OR perhaps I misunderstood and both are perfectly reversible? If so, that is worth elaborating on.

Regarding the second part on diffusion and moving boundaries, I would think the manuscript needs a better argumentation for novelty. For instance, the paragraph on wave pinning (now on page 10) could be moved forward to better provide a contrast between the author's work and existing literature. (Just a suggestion!) Also, some relevant literature was missed (understandably, it is not easy to find): "Having the right proportions: Interacting interfaces ensure robust spatial patterning" Vakulenko et al., Physical review letters 103 (16), 2009.

The diffusion mechanism seems fragile to me; slight changes in initial conditions appear to stongly affect the dynamics of boundary formation (Figure 3a). Since patterns are often continuously interpreted by downstream genes, this can have profound effects on the developmental process. I guess the issue is that in development most patterns are built on top of already existing patterns, whilst here the point is made that patterns even arise from uniformly expressed genes. So some asymmetry in the initial distribution of u1 and u2 (a pre-pattern) should make the dynamics very stable. Is that correct? This point is probably worth a paragraph in the manuscript.

Minor comments:

- The liberal use of quotation marks to mark special words, breaks my reading flow. I would in many cases simply remove them.

- Fig 1b: the gray gradients are difficult to compare. Could you plot them all in each graph, but the focal one darker and the other light-gray?

- Fig 2: why do the red and golden curves have a bit of a bump just before the gradient?

- Fig 2: it is not always clear which color is u1/u2. Repeat the legend of fig 1? Also the little network of Bcd and Hb could have colors corresponding with the graph below.

- Fig 3a: a legend for the temporal color code is missing. (dark gold to light gold)

**Have all data underlying the figures and results presented in the manuscript been provided?**

Reviewer #1: Yes

Reviewer #2: Yes

Reviewer #3: Yes

Reviewer #4: None

PLOS authors have the option to publish the peer review history of their article (what does this mean?). If published, this will include your full peer review and any attached files.

Reviewer #1: No

Reviewer #2: **Yes: **Timothy Saunders

Reviewer #3: No

Reviewer #4: No
---

## [Decision Letter · Decision Letter 1]

8 Apr 2021

Dear Dr. Perkins,

Thank you very much for submitting your manuscript "Dynamic positioning and precision of bistable gene expression boundaries driven by morphogen gradients" for consideration at PLOS Computational Biology. As with all papers reviewed by the journal, your manuscript was reviewed by members of the editorial board and by several independent reviewers. The reviewers appreciated the attention to an important topic. Based on the reviews, we are likely to accept this manuscript for publication, providing that you modify the manuscript according to the review recommendations.

Please address the major concerns of Reviewer 1 by updating the introduction and discussion. Also consider updating the ttitle as well.

Sincerely,

Attila Csikász-Nagy

Associate Editor

PLOS Computational Biology

Florian Markowetz

Deputy Editor

PLOS Computational Biology

[LINK]

Reviewer's Responses to Questions

**Comments to the Authors:**

Reviewer #1: Even though the author has worked hard into creating a new version of the manuscript (it actually looks like a fully new manuscript), I still hold the same main concerns I posed in the first version. The paper is formed by two disjointed parts that do not provide deep enough - biologically relevant - analysis by themselves. I still believe that the author should split the article in two and address the biological problems/limitations that barely mentions in the discussion. In the same line, the new title of the paper seems too generic, it does not address the main two topics that the paper tackles 1) dynamic morphogen (different to "dynamic positioning") and 2) diffusion of gene products. Two of my main concerns that are still present in the paper are:

1) On one hand the use of arbitrary random initial conditions while ignoring noise in the evolution. If the evolution of the systems is considered to be deterministic, where does the noise in initial conditions come from? Without a more dedicated analysis, it looks like the author is cherry-picking the biological conditions of the system. This should not only be mentioned in the discussion, but should be addressed in the introduction, and also tested along the manuscript. The author is aware of this since makes references to papers that have addressed these issues in the past, so I found misleading that this was not addressed properly along the text. For instance before starting to present their theory, the authors should make clear what other approaches have been done in the past regarding moving boundaries (e.g. refs [15] and [29]), and what place does their study plays in it.

2) The PDE/ODE travelling fronts still lacks an explicit connection between theory and results. In the text is not clear at all the connection between (13) and (14) or how it is applied in the Hb-Bcd case. There is a bit more of insight in the Appendix, but still is very hard to follow, or to understand what exactly do we learn out of it or how should we apply it. This would be more clear in an individual manuscript where the motivation, previous work, and results are well structured.

Reviewer #2: The author has done a very good job dealing with the range of reviewer queries. The paper is much improved and more rigorous.

Reviewer #3: This revision has clarified all my concerns and I would recommend its publication.

**Have all data underlying the figures and results presented in the manuscript been provided?**

Reviewer #2: Yes

PLOS authors have the option to publish the peer review history of their article (what does this mean?). If published, this will include your full peer review and any attached files.

Reviewer #1: No

Reviewer #2: No

Reviewer #3: No

**Have the authors made all data and (if applicable) computational code underlying the findings in their manuscript fully available?**

Reviewer #1: Yes

Reviewer #3: Yes

Figure Files:

Data Requirements:

Reproducibility:

References:

---

## [Editor Report · Decision Letter 2]

11 May 2021

Dear Dr. Perkins,

We are pleased to inform you that your manuscript 'Implications of diffusion and time-varying morphogen gradients for the dynamic positioning and precision of bistable gene expression boundaries' has been provisionally accepted for publication in PLOS Computational Biology.

Best regards,

Attila Csikász-Nagy

Associate Editor

PLOS Computational Biology

Florian Markowetz

Deputy Editor

PLOS Computational Biology

---

## [Editor Report · Acceptance letter]

25 May 2021

PCOMPBIOL-D-20-02221R2 

Implications of diffusion and time-varying morphogen gradients for the dynamic positioning and precision of bistable gene expression boundaries

Dear Dr Perkins,

I am pleased to inform you that your manuscript has been formally accepted for publication in PLOS Computational Biology. Your manuscript is now with our production department and you will be notified of the publication date in due course.

With kind regards,

Katalin Szabo
